# Dose–Response Curves of Pelargonic Acid against Summer and Winter Weeds in Central Italy

**Euro Pannacci \*** , **Daniele Ottavini, Andrea Onofri** and **Francesco Tei**

Department of Agricultural, Food and Environmental Sciences, University of Perugia, Borgo XX Giugno, 74-06121 Perugia, Italy
\* Correspondence: euro.pannacci@unipg.it

**Abstract:** Pelargonic acid is a non-selective post-emergence contact bio-herbicide which is registered both for cropping and non-cropping uses in several countries. Dose–response curves on the efficacy of pelargonic acid against common weeds in Mediterranean areas are not available. Dose–response curves of pelargonic acid efficacy against summer and winter annual weeds were evaluated in two field experiments (winter exp. in 2019 and summer exp. in 2020) in central Italy. Pelargonic acid was applied at five doses (1.4, 2.7, 5.4, 10.9 and 21.8 kg a.i. ha$^{-1}$). Data on weed density, weed dry weight, and weed ground cover were used to calculate the efficacy of pelargonic acid against winter and summer weeds. Data were subjected to a non-linear regression analysis using the logistic dose–response model. Dose of pelargonic acid required to obtain 50%, 70%, 90% and 95% weed control against each weed species ($ED_{50}$, $ED_{70}$, $ED_{90}$ and $ED_{95}$) were estimated. ED values allowed us to classify winter and summer weeds with respect to their susceptibility to pelargonic acid ($ED_{50}$ values in kg ha$^{-1}$ are reported in parenthesis): *Kickxia spuria* (2.6) (more susceptible) > *Heliotropium europaeum* (3.0) > *Echinochloa crus-galli* (3.4) > *Solanum nigrum* (3.6) > *Stachys annua* (5.3) > *Papaver rhoeas* (6.5) > *Veronica hederifolia* (10.3) > *Amaranthus retroflexus* (11.4) > *Matricaria chamomilla* (11.6) > *Portulaca oleracea* (18.7) > *Lolium multiflorum* (>21.8) (less susceptible). These findings will allow for the optimization of weed control by pelargonic acid and its use in weed management strategies, both in organic and sustainable cropping systems, under different environmental conditions.

**Keywords:** bioherbicides; weed control; natural product; effective dose; integrated weed management

## 1. Introduction

### 1.1. Sustainable Weed Management

Sustainable weed management requires an effective understanding of the biology of weed species and the possible impacts of practices on changes in weed community and individual weed species evolution. Herbicides have been the dominant tool for weed management in agricultural crops, horticulture, and amenity turfgrass [1]. Synthetic herbicides allow for effective and cheap weed control, although their use is currently highly debated due to downsides such as the development of resistance, water contamination, reduction of biodiversity, and potential human toxicity [2]. Furthermore, the constraints of herbicide resistance, lack of new herbicide development, and greater regulatory scrutiny could continue to erode the use of herbicides, as well as other pesticides [3]. Therefore, a combined use of different weed control methods (cultural, physical, mechanical and chemical) is required in an integrated weed management system (IWMS), with the aim of preserving herbicide technology, to complement their efficacy, and decrease their use in agricultural systems [3–6].

### 1.2. Natural Herbicides and Pelargonic Acid

Among the most promising innovative tactics and technologies that could contribute to designing integrated weed management practices, are natural herbicides, often based on

terpene hydrocarbons or oxygenated compounds or fatty acids [7–11]. Pannacci et al. [7,9] found that the *Artemisia vulgaris* L. aerial biomass extract (mainly composed by seven hydroxycinnamic acids and five hydroxybenzoic acids) at 25% *w/v* concentration proved to be a potent inhibitor of seed germination and plant growth of the weeds *Lolium multiflorum* Lam. and *Amaranthus retroflexus* L, without affecting seed germination and growth of the crops *Triticum aestivum* L. and *Zea mays* L. The efficacy against weeds, and selectivity to the crops, are two important qualities from which the development of the aqueous extract of *A. vulgaris* as a pre-emergence bioherbicide could be considered. Furthermore, from an agronomic point of view, the use of water extracts from bioactive plants, such as *A. vulgaris* extract, is economically viable for farmers, especially those living in poor communities, since they would only need to source the plants and mix them with water, which is inexpensive and does not require any specialist knowledge [9]. Furthermore, water extracts have many advantages, such as being eco-friendly, easily degradable, and not being persistent in the soil. Nor is it toxic to animals and humans. It can reduce the use of chemical herbicides [10]. Sorghum extract is one such natural weed inhibitor, providing soluble allelochemicals phytotoxic to certain weeds. *Ocimum basilicum* L. extract can also be applied as a biodegradable herbicide due to the content of allelochemicals, whereas essential oils extracted from Lemon eucalyptus (*Eucalyptus citriodora*) and field mint (*Mentha arvensis*) have demonstrated phytotoxicity to weeds [10]. However, natural compounds could have negative characteristics compared to synthetic compounds, e.g., lower effectiveness, lack of persistence, and inability to reach and penetrate the target plant. Thus, recently, certain nanomaterials have been explored that could facilitate the development of formulated natural pesticides, making them more effective and more environmentally friendly. Nano-formulations can improve efficacy, reduce effective doses, and increase shelf-life and persistence [11].

Fatty acids are non-systemic contact herbicides that solubilize lipids, affecting the cuticle which protects the leaf from evaporation and desiccation. When sprayed on foliage, fatty acids cause wilting, and then death. For that reason, these products can be used for vegetation desiccation [2,12]. Some fatty acid salts are now marketed as non-selective herbicidal "soaps." These are composed of aliphatic fatty acids of varying lengths mixed with vinegar or acetic acid and emulsifiers. They act as disinfectants in a relatively short time and have no selectivity, but most of the weeds tend to recover because there is no residual activity. The persistence of these natural products is generally lower, and as a consequence, their impact on the environment is also smaller [13]. Among the fatty acids, the most representative is pelargonic acid, which is currently integrated into different marketed bioherbicides [14]. Pelargonic acid, extracted for the first time from *Pelargonium roseum* Willd leaves, is a non-selective post-emergence contact bioherbicide for the control of annual broad-leaved and grass weeds, which is registered both for cropping and non-cropping uses in several countries. It destroys cell membranes, causing a rapid loss of cellular functions. It has only contact activity, and, thus, it is necessary to spray most of the plant for complete control. Being environmentally benign, it is suitable for high-traffic areas such as paths and recreational turfs, especially if these are also frequented by pets or children; however, the high cost of these products is a major deterrent for adoption [15]. In cropping systems, pelargonic acid can be mainly used to control annual weeds using the false seedbed technique as an alternative to nonselective synthetic herbicides, whose use has been restricted or might become more restricted [16].

### 1.3. Objective, Novelty and Importance of the Study

As opposed to synthetic herbicides, the available natural herbicides, such as pelargonic acid, have little to no selectivity and they must be applied in relatively large quantities [16]. Furthermore, there is little available in the scientific literature on the use and environmental impact of natural products in sustainable agriculture. In fact, that the bioherbicides are naturally derived does not mean they are actually harmless, and, in any case, an optimization of their use is needed in order to increase their efficacy, reducing their potential side

effects [14,15]. In an Integrated Weed Management System (IWMS), an effective way to reduce the side effects of the herbicides, but also applied to the bioherbicides, is to apply the lowest dose needed for biologically effective weed control [17]. Several studies have demonstrated satisfactory weed control and acceptable crop yields when herbicides are used at lower than normally recommended doses [17,18]. Application doses of post-emergence herbicides can indeed, be substantially reduced if the minimum dose requirement for satisfactory efficacy (i.e., Effective Dose, $ED_{90}$ and $ED_{95}$, the doses required to obtain 90% and 95% weed control) is known with respect to the most common "herbicide–weed species" combinations [18]. Furthermore, the knowledge of $ED_{90}$ and $ED_{95}$ values is one of the main factors in the implementation of the Decision Support Systems (DSS) for Integrated Weed Management with the aim to decrease the dependence on herbicides, that is one of the main objectives within the EU Green Deal. The determination of ED values requires dose–response studies for each "herbicide-weed species" under various environmental conditions, that allows, also, to define the sensitiveness of different weed species to an herbicide [18]. In this context, the data of pelargonic acid efficacy against common weeds are not broadly available and normally referred to the effects at maximum labelled dose, whereas not data are available on dose–response curves of pelargonic acid for mixed weed stands of weeds common in Europe and Mediterranean areas [2,10,16]. For these reasons, the objective of this study was to determine dose–response curves of pelargonic acid efficacy against summer and winter annual weeds common in central Italy and the Mediterranean area.

## 2. Materials and Methods

### 2.1. Field Experiments: Site, Description and Treatments

Two field experiments were carried out in 2019 (winter) and 2020 (summer) at the Experimental Station of the Department of Agricultural, Food and Environmental Sciences—University of Perugia, located in the middle of the Tiber plain (Central Italy, 42.96° N, 12.37° E, 165 m a.s.l.), in a field with a clay-loam soil (25% sand, 30% clay and 45% silt, pH 8.2, 0.9% organic matter).

False seedbed preparation was carried out in the field, both in the winter experiment (18 November 2019) and the summer experiment (29 May 2020), with the aim of stimulating the weeds' emergence due to the natural weed seed bank. Four weeks after false seedbed preparation (18 December 2019 in winter exp. and 29 June 2020 in summer exp.), pelargonic acid (Beloukha, 680 g a.i. $L^{-1}$, maximum labelled dose: 21.8 kg a.i. $ha^{-1}$, Jade, France) was applied at five doses (1.4, 2.7, 5.4, 10.9 and 21.8 kg a.i. $ha^{-1}$), using a backpack plot sprayer calibrated to deliver 300 L $ha^{-1}$ spray liquid at 200 kPa, and with the weeds at the growth stages (according BBCH-scale) [19] reported in Table 1. Untreated plots were added as controls. The experimental design was a randomized block with four replicates and a plot size of 11.25 $m^2$ (5 m × 2.25 m).

### 2.2. Measurements

In each experiment, four weeks after the treatments (WAT), weed ground cover (%) was rated visually on the central part of each plot, using the Braun–Blanquet cover-abundance scale [20]. Furthermore, in the winter experiment, weeds on two quadrats (0.5 × 0.5 m each one) per plot were collected, counted, weighed, and oven-dried at 105 °C to evaluate weed density and weed dry weight in treated and untreated plots, whereas, in the summer experiment weed density and weed biomass were not evaluated, due to COVID-19 restrictions.

Meteorological data (daily maximum and minimum air temperature and rainfall) were collected from a nearby weather station. Ten days mean of daily values were calculated and compared with multi-annual average values (from 1921) (Figure 1).

**Table 1.** Weed species in the untreated control of the two experiments: BBCH growth stages at treatments time and weed parameters evaluated 4 weeks after treatments (WAT). Standard errors are in parentheses.

| Weeds Species | BBCH-Scale (at Treatments Time) | 4 WAT | | | | | |
|---|---|---|---|---|---|---|---|
| | | Ground Cover (%) | | Density (n. Plants m⁻²) | | Dry Biomass (g m⁻²) | |
| Winter experiment | | | | | | | |
| *Papaver rhoeas* | 14–18 | 17.5 | *(0.2)* | 155.3 | *(17.5)* | 14.1 | *(1.6)* |
| *Matricaria chamomilla* | 14–18 | 19.4 | *(1.9)* | 109.3 | *(18.4)* | 3.2 | *(1.1)* |
| *Lolium multiflorum* | 13–21 | 9.7 | *(3.0)* | 35.3 | *(15.7)* | 4.0 | *(2.4)* |
| *Veronica hederifolia* | 14–22 | 21.3 | *(2.2)* | 93.3 | *(32.3)* | 10.3 | *(0.6)* |
| Total | | 67.8 | *(6.0)* | 393.3 | *(50.7)* | 31.6 | *(1.9)* |
| Summer experiment | | | | | | | |
| *Stachys annua* | 18–22 | 5 | *(0.1)* | - | | - | |
| *Amaranthus retroflexus* | 14–22 | 50 | *(0.1)* | - | | - | |
| *Portulaca oleracea* | 14–24 | 84 | *(3.1)* | - | | - | |
| *Echinochloa crus-galli* | 21–22 | 18 | *(0.1)* | - | | - | |
| *Kickxia spuria* | 16–18 | 5 | *(0.1)* | - | | - | |
| *Solanum nigrum* | 12–14 | 18 | *(0.1)* | - | | - | |
| *Heliotropium europaeum* | 12–14 | 5 | *(0.1)* | - | | - | |
| Total | | 184.4 | *(3.3)* | - | | - | |

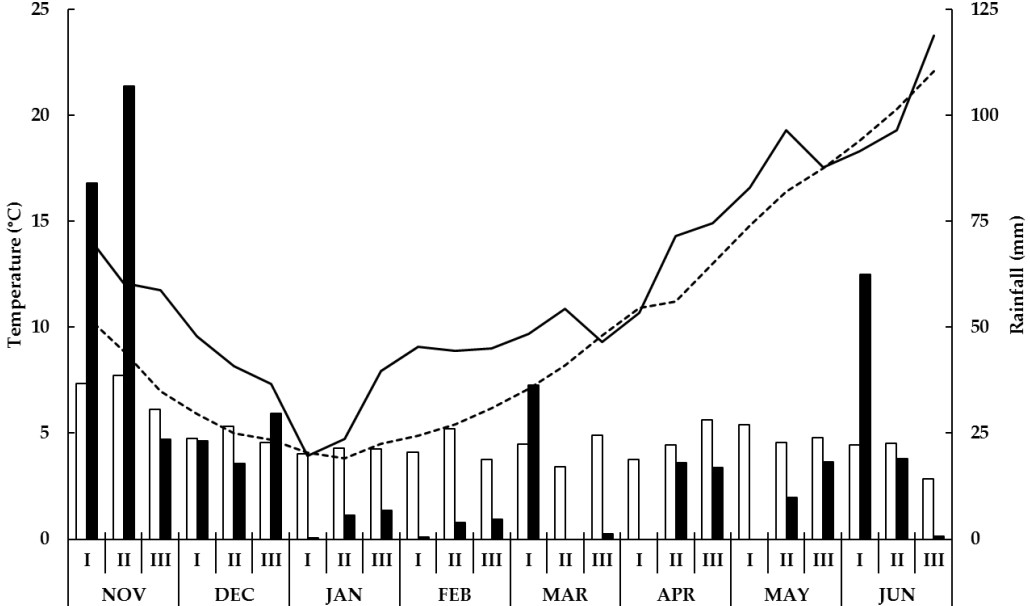

**Figure 1.** Ten-day mean rainfall (bold bar) and ten-day mean air temperature (continuous line) during the periods of the two experiments (November 2019–June 2020), compared to the ten-day rainfall (empty bar) and ten-day mean air temperature (dashed line) as a multi-annual average (from 1921).

Data on weed density, weed dry weight, and weed ground cover were used to calculate the efficacy (*E*) of different pelargonic acid treatments relative to the untreated control, according to the following equation, as was also reported in Pannacci et al. [21]:

$$E(\%) = \frac{W_U - W_T}{W_U} \times 100 \qquad (1)$$

where,
$W_U$: weed density or weed dry weight or weed ground cover in untreated plots.
$W_T$: weed density or dry weight or weed ground cover in treated plots.

### 2.3. Statistical Analysis

For each "weed-pelargonic acid" combination, the data of pelargonic acid efficacy were subjected to a non-linear regression analysis using the following logistic dose–response model [22]:

$$Y = \frac{100}{1 + exp\{-b[\log(x) - \log(ED_{50})]\}} \tag{2}$$

where $Y$ is the percentage efficacy of pelargonic acid against each weed, $x$ is the dose of pelargonic acid, $b$ is the slope of the curve around the inflection point, $ED_{50}$ is the dose required to give 50% weed control.

The $ED_{50}$ parameter can be replaced by any $ED$ level, so the selected model was used to estimate the dose of pelargonic acid required to obtain 70%, 90% and 95% weed control against each weed species, ($ED_{70}$, $ED_{90}$ and $ED_{95}$ values) [17,18,23]. When the upper asymptote did not reach 100%, it was included as a parameter in the model fitting [22].

The logistic dose–response model was directly fitted to the experimental data, by using the EXCEL® Add-in macro BIOASSAY97 [24]. The goodness-of-fit was assessed by graphical analyses of residuals and F-test for lack-of-fit [25].

## 3. Results and Discussion

### 3.1. Weed Species and Dose–Response Curves in the Two Experiments

The two experiments were characterized by a different weed flora composition (Table 1). In the winter experiment, the weed species in the untreated control were *Papaver rhoeas* L., *Matricaria chamomilla* L., *Veronica hederifolia* L. and *L. multiflorum*, listed in descending order of density that showed high values (393 plants m$^{-2}$, as total weed density) at 4 WAT (Table 1). On the contrary, ground cover and dry biomass values were low due to a slow growth of weeds under decreasing air temperature conditions from the treatment time (18 December 2019) to 4 WAT (Figure 1). The growth stages of the winter weeds at the moment of treatments were similar, ranging from 14 to 18 or 22 BBCH scale for the dicotyledonous weeds and from 13 to 21 for the grass weed (*L. multiflorum*) (Table 1).

In the summer experiment the weed species in the untreated control were, mainly, *Portulaca oleracea* L, *A. retroflexus*, *Solanum nigrum* L, *Echinochloa crus-galli* (L.) P. Beauv., listed in descending order of ground cover, plus other less abundant species (5% of ground cover), such as *Stachys annua* L., *Kickxia spuria* (L.) Dumort. and *Heliotropium europaeum* L. (Table 1). The growth stages of the summer weeds, at the moment of treatments, ranged from 12–14 to 22–24 BBCH scale for the dicotyledonous weeds and 21–22 for the grass weed (*E. crus-galli*) (Table 1).

Dose–response curves for pelargonic acid against winter and summer weed species always showed a good fit to the experimental data (Figures 2 and 3, respectively). $ED_{50}$, $ED_{70}$, $ED_{90}$ and $ED_{95}$ values for pelargonic acid and $b$ values are reported in Tables 2 and 3. These results show that the use of dose–response curves in field studies can be an important tool to estimate parameters which are biologically meaningful to express herbicide efficacy in terms of biological equivalent doses (ED values) [26,27]. Furthermore, the knowledge of $ED_{90}$ and $ED_{95}$ values for each weed species allows for the optimization of pelargonic acid application, increasing the efficacy of treatments and avoiding the maximum labeled dose. The ED values can also be implemented in the Decision Support Systems (DSS) for Integrated Weed Management with the aim of reducing the dependence on herbicides, one of the main objectives within the EU Green Deal.

### 3.2. Winter Weeds

Concerning winter weeds, ED values of each species were very similar among the different assessed parameters of weeds (ground cover, density and biomass) (Table 2), confirming that a subjective visual assessment, such as weed ground cover (by the Braun–Blanquet cover abundance scale), can have the same reliability and accuracy as objective assessments (weed density and biomass), if it can be referred to an untreated control to relativize the assessments [28,29].

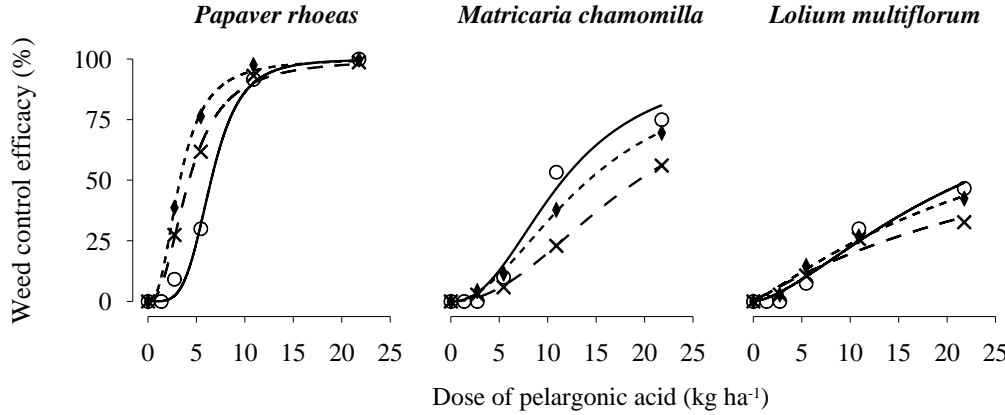

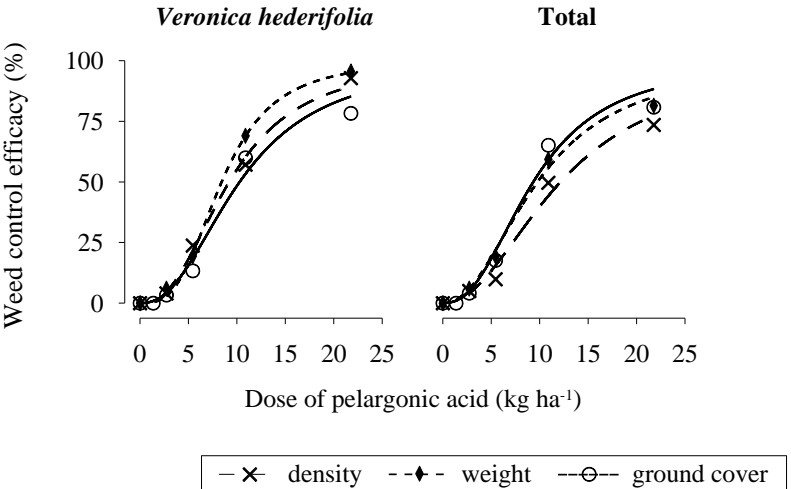

density    weight    ground cover

**Figure 2.** Dose-response curves for the efficacy of pelargonic acid against winter weed species based on their density, weight, and ground cover data. Symbols show observed data and lines show fitted curves according to Model [2].

In particular, *P. rhoeas* was the most sensitive to pelargonic acid, as in shown in the dose–response curves, and the lowest ED values (Figure 2; Table 2). In more detail, this weed could be well controlled (with an efficacy of 90% or 95%) using doses of pelargonic acid from 10 to 15 kg a.i. ha$^{-1}$ (Table 2). *V. hederifolia* was less sensitive to pelargonic acid than *P. rhoeas*, with the ED$_{90}$ and ED$_{95}$ values, in terms of weed biomass control, of 17.1 and 21.7 kg a.i. ha$^{-1}$, respectively; whereas ED$_{90}$ and ED$_{95}$ values for weed ground cover and weed density were up of the maximum labelled dose (>21.8 kg a.i. ha$^{-1}$) (Table 2). *M. chamomilla* was not well controlled (efficacy higher than 90%), regardless of pelargonic acid dose (ED$_{90}$ and ED$_{95}$ values > 21.8 kg a.i. ha$^{-1}$), whereas 70% of control can be obtained with dose of pelargonic acid lower than the maximum labelled dose (ED$_{70}$ of 16.8 and 21.7 kg a.i. ha$^{-1}$ in terms of weed ground cover and weed biomass, respectively) (Table 2 and Figure 4). *L. multiflorum* showed all ED levels higher than the maximum labelled dose, indicating a high tolerance of this weed to pelargonic acid (Figure 4).

These findings are consistent with those obtained by Ogbangwor and Söchting [30] who found that pelargonic acid at 31 kg a.i. ha$^{-1}$, caused, 3 WAT, a fresh weight reduction of 38% in *L. multiflorum* and 100% in *P. rhoeas*, *V. hederifolia* and *Matricaria matricaroides* auct., when treated at the BBCH 12 stage. Furthermore, it has been confirmed that the approach based on dose–response curves is able to quantify, by ED values, weed susceptibility to an herbicide, allowing for the classification of weeds with respect to their susceptibility to pelargonic acid. The ranking among winter weed species based on their susceptibility to pelargonic acid is: *P. rhoeas* (more susceptible) > *V. hederifolia* > *M. chamomilla* > *L. mul-*

*tiflorum* (less susceptible); confirming that dicotyledonous weed control was superior to monocotyledonous weed control when using a contact herbicide such as pelargonic acid (Figure 4) [30]. In fact, pelargonic acid consistently controlled dicotyledonous weeds more effectively than monocotyledonous weeds, due to the greater exposure of dicotyledonous weeds meristematic tissue (apical meristem) compared to grass plants' intercalary meristematic tissue and basal meristematic tissue (basal buds), from which the development of new shoots can be observed. Travlos et al. [31] found a 30% dry weight reduction in *L. rigidum* at seven days after a treatment with pelargonic acid applied at 10 kg a.i. ha$^{-1}$, that was similar to our percentage control of *L. multiflorum* based on dry weight (24%, data not shown but extrapolated by the relative dose–response curve) at the same dose (Figure 2).

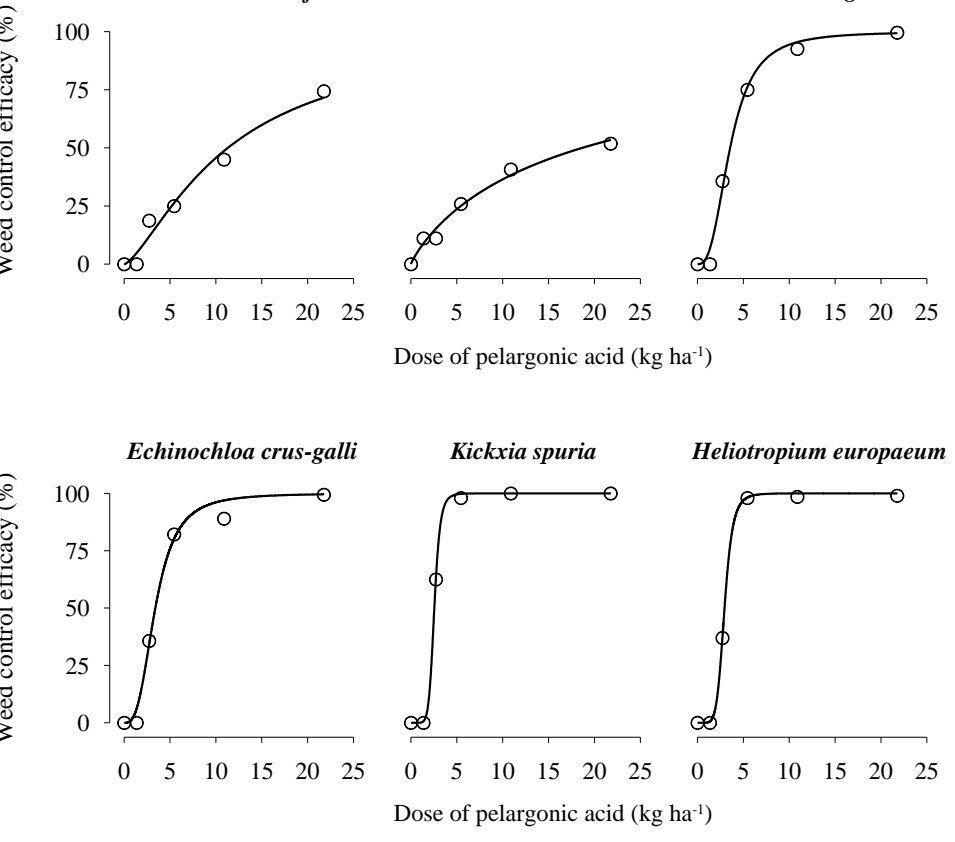

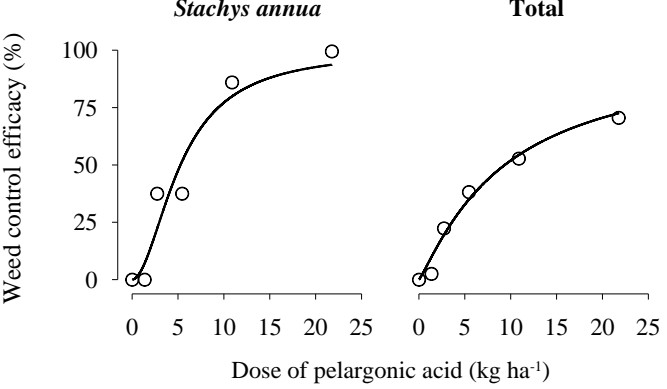

**Figure 3.** Dose-response curves for the efficacy of pelargonic acid against summer weed species based on their ground cover data. Symbols show observed data and lines show fitted curves according to Model [2].

**Table 2.** Dose-response curves parameters (b, ED$_{50}$, ED$_{70}$, ED$_{90}$, ED$_{95}$), based on weed ground cover, weed density, and weed biomass, calculated from the fitted equations in Figure 1. Standard errors are in parentheses.

| Weeds | b | | ED$_{50}$ (kg ha$^{-1}$) | | ED$_{70}$ (kg ha$^{-1}$) | | ED$_{90}$ (kg ha$^{-1}$) | | ED$_{95}$ (kg ha$^{-1}$) | |
|---|---|---|---|---|---|---|---|---|---|---|
| **Weed Ground Cover (%)** | | | | | | | | | | |
| *Papaver rhoeas* | 4.30 | (0.32) | 6.5 | (0.1) | 7.9 | (0.2) | 10.9 | (0.52) | 12.9 | (0.8) |
| *Matricaria chamomilla* | 2.29 | (0.21) | 11.6 | (0.5) | 16.8 | (1.0) | >21.8 | | >21.8 | |
| *Lolium multiflorum* | 1.54 | (0.14) | >21.8 | | >21.8 | | >21.8 | | >21.8 | |
| *Veronica hederifolia* | 2.34 | (0.27) | 10.3 | (0.5) | 14.8 | (1.0) | >21.8 | | >21.8 | |
| Total | 2.39 | (0.20) | 9.3 | (0.4) | 13.3 | (0.6) | >21.8 | | >21.8 | |
| **Weed density (n. plants m$^{-2}$)** | | | | | | | | | | |
| *Papaver rhoeas* | 2.36 | (0.32) | 4.2 | (0.1) | 6.1 | (0.2) | 10.8 | (0.8) | 14.8 | (1.4) |
| *Matricaria chamomilla* | 2.10 | (0.06) | 19.4 | (0.3) | >21.8 | | >21.8 | | >21.8 | |
| *Lolium multiflorum* | 1.01 | (0.21) | >21.8 | | >21.8 | | >21.8 | | >21.8 | |
| *Veronica hederifolia* | 2.47 | (0.22) | 9.2 | (0.4) | 13.0 | (0.6) | >21.8 | | >21.8 | |
| Total | 2.11 | (0.32) | 12.2 | (0.9) | 18.2 | (1.8) | >21.8 | | >21.8 | |
| **Weed biomass (g m$^{-2}$)** | | | | | | | | | | |
| *Papaver rhoeas* | 2.50 | (0.15) | 3.3 | (0.1) | 4.6 | (0.1) | 7.9 | (0.4) | 10.7 | (0.8) |
| *Matricaria chamomilla* | 2.00 | (0.07) | 14.2 | (0.2) | 21.7 | (0.5) | >21.8 | | >21.8 | |
| *Lolium multiflorum* | 1.17 | (0.13) | >21.8 | | >21.8 | | >21.8 | | >21.8 | |
| *Veronica hederifolia* | 3.11 | (0.18) | 8.4 | (0.2) | 11.1 | (0.3) | 17.1 | (0.8) | 21.7 | (1.3) |
| Total | 2.16 | (0.20) | 9.8 | (0.4) | 14.5 | (0.9) | >21.8 | | >21.8 | |

**Table 3.** Dose-response curves parameters (b, ED$_{50}$, ED$_{70}$, ED$_{90}$, ED$_{95}$), based on weed ground cover, calculated from the fitted equations in Figure 2. Standard errors are in parentheses.

| Weeds | b | | ED$_{50}$ (kg ha$^{-1}$) | | ED$_{70}$ (kg ha$^{-1}$) | | ED$_{90}$ (kg ha$^{-1}$) | | ED$_{95}$ (kg ha$^{-1}$) | |
|---|---|---|---|---|---|---|---|---|---|---|
| **Weed Ground Cover (%)** | | | | | | | | | | |
| *Stachys annua* | 1.90 | (0.46) | 5.3 | (0.7) | 8.2 | (1.4) | 16.8 | (5.2) | >21.8 | |
| *Amaranthus retroflexus* | 1.41 | (0.19) | 11.4 | (1.0) | 20.7 | (2.7) | >21.8 | | >21.8 | |
| *Portulaca oleracea* | 0.89 | (0.06) | 18.7 | (1.2) | >21.8 | | >21.8 | | >21.8 | |
| *Echinochloa crus-galli* | 2.97 | (0.28) | 3.4 | (0.1) | 4.5 | (0.2) | 7.1 | (0.6) | 9.2 | (0.9) |
| *Kickxia spuria* | 8.01 | (1.99) | 2.6 | (0.04) | 2.8 | (0.03) | 3.4 | (0.2) | 3.7 | (0.3) |
| *Solanum nigrum* | 2.73 | (0.21) | 3.6 | (0.1) | 4.9 | (0.2) | 8.0 | (0.6) | 10.5 | (0.9) |
| *Heliotropium europaeum* | 6.53 | (0.73) | 3.0 | (0.03) | 3.4 | (0.1) | 4.1 | (0.2) | 4.6 | (0.3) |
| Total | 1.18 | (0.08) | 9.0 | (0.5) | 18.5 | (1.4) | >21.8 | | >21.8 | |

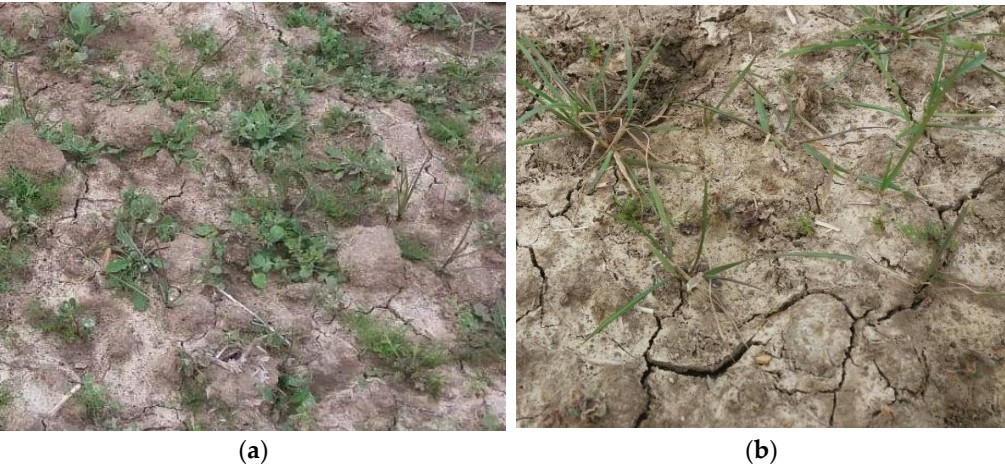

(**a**)                    (**b**)

**Figure 4.** Weeds in the winter experimental plots at 4 WAT: (**a**) untreated control; and (**b**) pelargonic acid at maximum dose (21.8 kg a.i. ha$^{-1}$, see the less susceptible weeds: *L. multiflorum* and *M. chamomilla*).

### 3.3. Summer Weeds

Concerning summer weeds, ED values were always lower than maximum labelled rate, except for *S. annua* (ED$_{95}$), *A. retroflexus* (ED$_{90}$ and ED$_{95}$) and *P. oleracea* (ED$_{70}$, ED$_{90}$ and ED$_{95}$) that showed to be the less susceptible weeds to phytotoxic effects of pelargonic acid (Figure 5). Covarelli and Contemori [32] found that pelargonic acid at 30 kg a.i. ha$^{-1}$ was able to control *P. oleracea* at cotyledons growth stage (BBCH 10–12), while when applied with the weed at the late growth stages (plant diameter ranged from 5 to 40 cm, corresponding to BBCH from 18 to 25), the efficacy decreased to 30%, that is in line with our results. In contrast with our results, the same authors found an efficacy of 100% of pelargonic acid against *A. retroflexus* at late growth stages (BBCH 18-20), as well as Muñoz et al. [33] that observed a 100% efficacy of pelargonic acid (applied at 5%, 8% and 10% concentrations rate) against *A. retroflexus* treated at early growth stage (BBCH 14). The low efficacy of pelargonic acid against *A. retroflexus* in our experiment could be due to the high growth stage of this weed at the treatment time and to the lower dose of pelargonic acid with respect to that used in the referred experiments. This explanation can be supported by Ogbangwor and Söchting [30] who concluded that, in order to achieve stronger weed control, pelargonic acid will need to be sprayed on smaller, younger weeds or at a higher application rate. In our experiment, the dose–response curve of *A. retroflexus* showed an efficacy of 74% at the highest dose of pelargonic acid (21.8 kg a.i. ha$^{-1}$) (Figure 3), but the efficacy should increase upon increasing the dose of pelargonic acid.

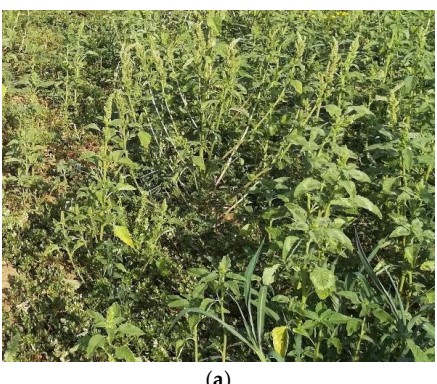 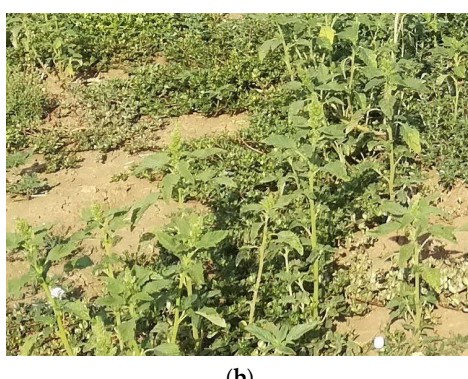

(**a**)          (**b**)

**Figure 5.** Weeds in the summer experimental plots at 4 WAT: (**a**) untreated control; and (**b**) pelargonic acid at a maximum dose (21.8 kg a.i. ha$^{-1}$, see the less susceptible weeds: *P. oleracea* and *A. retroflexus*).

The other summer weeds were more susceptible than previous weeds to the phytotoxic effects of pelargonic acid, showing ED$_{95}$ ranging from 3.7 kg a.i. ha$^{-1}$ in *K. spuria* to 10.5 kg a.i. ha$^{-1}$ in *S. nigrum*. The ranking among summer weed species based on their susceptibility to pelargonic acid is: *K. spuria* (more susceptible) > *H. europaeum* > *E. crus-galli* > *S. nigrum* > *S. annua* > *A. retroflexus* > *P. oleracea* (less susceptible). Concerning grass weeds, *E. crus-galli* was very susceptible to the phytotoxicity of pelargonic acid with ED$_{90}$ and ED$_{95}$ of 7.1 and 9.2 kg a.i. ha$^{-1}$, respectively, despite the high growth stage of the weeds. These results are in full agreement with previous studies showing that pelargonic acid can achieve a high efficacy against *E. crus-galli*, when also applied at late growth stages [30,32,34].

### 3.4. Total Weeds and Pelargonic Acid Considerations

In both experiments, the weed control efficacy against the total weeds was affected by the susceptibility to pelargonic acid of the most abundant weeds, with ED$_{90}$ and ED$_{95}$ values always over the highest dose, at which total weed control of 71% and 73% were achieved for the summer and winter experiment, respectively (Figures 2 and 3; Tables 2 and 3). To overcome these limits when trying to increase the weed control, our and previous results seem to suggest that applications of pelargonic acid at high concentrations targeting small and younger weeds, especially in the presence of unsusceptible weeds, with the

possibility to repeat treatment if needed, could be a good strategy for effective weed control [2,34,35]. This is especially noted when applications are conducted under field conditions, as revealed by Kanatas et al. [36] who found that applying pelargonic acid twice within a two-week interval can improve the herbicidal effects on *E. crus-galli* and common broadleaved weed species, such as *Chenopodium album* L., *S. nigrum*, *A. retroflexus* and *Mercurialis annua* L. In this context, pelargonic acid is also a promising alternative herbicide that can be implemented in integrated weed management systems (IWMS) and can be combined with the use of other practices [34]. For instance, in cropping systems, it can be mainly used to control annual weeds using the false seedbed technique as an alternative to nonselective synthetic herbicides. In particular, the study showed that a stale seedbed combined with pelargonic acid applications, reduced annual weed density by 95% as compared to a normal seedbed, indicating that such pelargonic acid-based herbicides can be equally effective as glyphosate against annual weeds in a stale seedbed where a crop is about to be established, and thus reap the benefits of pre-sowing weed elimination [37]. Such practices can also contribute to the development of more environmentally friendly weed management strategies, the reduction of herbicide inputs in agriculture so as to achieve European green deal goals, and the management of weed biotypes that have developed resistance to synthetic herbicides [34,38,39]. Furthermore, the addition of the commercial adjuvant used in the preparation of the spray solution can contribute to a further increase in the efficacy of pelargonic acid on the target weeds. Some research studies have found that the addition of diammonium succinate and succinic acid improved the efficacy of pelargonic acid from 117% to 200% under greenhouse conditions, while natural adjuvants (garlic extracts and yucca extracts) improved the efficacy of pelargonic acid [35,40]. In any case, considering the potential of adjuvants to improve the herbicidal activity of pelargonic acid, further adjuvants should be evaluated by conducting additional pot and field trials repeated in space and time [34].

## 4. Conclusions

To date, no studies have evaluated the dose–response curves of pelargonic acid efficacy against summer and winter annual weeds common in Italy and the Mediterranean area. The findings of the present study confirmed that the use of dose–response curves in field studies is an important tool for estimating the parameters which are biologically meaningful to express herbicide efficacy in terms of biological equivalent doses. In particular, the approach based on dose–response curves quantified, by ED values, the weed susceptibility to pelargonic acid, allowing for the classification of weeds with respect to their susceptibility to this herbicide. The ranking among winter and summer weed species based on their susceptibility to pelargonic acid, quantified by $ED_{50}$ of weed ground cover, was: *Kickxia spuria* (more susceptible) > *Heliotropium europaeum* > *Echinochloa crus-galli* > *Solanum nigrum* > *Stachys annua* > *Papaver rhoeas* > *Veronica hederifolia* > *Amaranthus retroflexus* > *Matricaria chamomilla* > *Portulaca oleracea* > *Lolium multiflorum* (less susceptible).

Further research is needed to evaluate more combinations of "weeds–pelargonic acid" by dose–response curves in order to increase the ED values data and optimize weed control by pelargonic acid and its use in weed management strategies, both in organic and sustainable cropping systems, under different environmental conditions.

**Author Contributions:** Conceptualization, E.P.; methodology, E.P.; data curation, E.P. and D.O.; formal analysis, E.P.; resources, E.P., A.O. and F.T.; writing—original draft preparation, E.P.; writing—review and editing, E.P., D.O., A.O. and F.T.; project administration, E.P.; funding acquisition, E.P., A.O. and F.T. All authors have read and agreed to the published version of the manuscript.

**Funding:** This research was partially funded by the project: "Fondo d'Ateneo per la Ricerca di Base 2017—(Riattivazione 2019)—Department of Agricultural, Food and Environmental Sciences, University of Perugia".

**Institutional Review Board Statement:** Not applicable.

**Informed Consent Statement:** Not applicable.

**Data Availability Statement:** Not applicable.

**Acknowledgments:** The authors would like to acknowledge Luchetti Daniele (Dept. of Agricultural, Food and Environmental Sciences, University of Perugia) for technical assistance in the field experiments.

**Conflicts of Interest:** The authors declare no conflict of interest.

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
