# Peer review of "Dose–Response Curves of Pelargonic Acid against Summer and Winter Weeds in Central Italy"

_agronomy, doi:10.3390/agronomy12123229_

Round 1
Reviewer 1 Report
I commend the authors Pannacci et al of the manuscript titled “Dose-response curves of pelargonic acid against summer and winter weeds” for their work on the evaluation of pelargonic acid efficiency as bio herbicide against winter and summer weeds in central Italy.
The manuscript is interesting, however, I have some points to be applied to enable this work to be published and increase its readership and acceptability:
1. In the abstract,
- Please include the values of (weed control efficiency and related dose of pelargonic acid) or the ED50% because it’s the main result you have
2. In the introduction part: I feel the introduction was written in hurry, please split the introduction into paragraphs, introductory paragraph, subject details (2 paragraphs) and last paragraph the objective and novelty and importance of the work.
It need to be reorganized.
3. In the materials and methods part:
- The material and methods part need to reorganized to subsections, showing plant material used, sources and details, treatments, measurement, statistical analyses. Its not readable in the current form.
- Where did you bring the seed? There is no information about that. The collection sites should be declared, the vouchers should be shown.
- Data about weed density, weed dry weight and weed ground cover measurements should be mentioned in detail here beside the citation.
- Additional description about The ED50 parameter should be added
4. In the results and discussion
o The other figures of the other species of the weeds response to pelargonic acid could be added as well, the data is very interesting.
o I see a lot of sources in literature are talking about the use of pelargonic acid as herbicide, please defend the novelty of the work, both in introduction and discussion.
I give you Major revision
Reviewer 2 Report
The paper titled : “Dose-response curves of pelargonic acid against summer and winter weeds”, submitted by the authors Euro Pannacci, Daniele Ottavini, Andrea Onofri and Francesco Tei investigates pelargonic acid effects as bioherbicide against specific weeds in Italy. The paper contains good amount of data about the selected topic and is if special interest for researchers within this field. There are some things need to be addressed before the publishing of this paper:
- You may add the location of the experiment in the title “Italy” or central Italy.
- The effective dose of the pelargonic acid need to be mentioned in the abstract
- The prospect of the work or the application of this study need to be added to the abstract
-
- The introduction is very short, and additional data need to be added about each studied weed and current knowledge about other bioherbicides used. Furthermore, previous work on pelargonic acid need to be mentioned more in detail and the novelty of the current investigation need to be highlighted. There are reviews about pelargonic acid as you see below:
https://www.sciencedirect.com/topics/agricultural-and-biological-sciences/pelargonic-acid
- Please use paragraphs in the introduction for clarity and orientation.
- In lines 196- 200 please add the new part found in your study.
- In lines 282-232, was this expiation first time reported in your study, this should be declared here.
- In the results and discussion part please use the numbering of subsections to enhance the readability and citation of your work, it cannot be group of paragraphs.
- You may split the results and discussion part into summer and winter weed for example.
- In the conclusion, you added the prospect of the work which could be reflected in the abstract as well.
Reviewer 3 Report
Well prepared paper. Needs only small corrections:
Line 74: (5 m x 2.25 m)
Lines 167, 169, 193, 205: Figure 4 or Photo 1?
Lines 217, 246: Figure 5 or Photo 2?
Line 213: (Figure 2).
Line 231: (Figure 3),
Line 252: (Figure 2 and 3; Table 2 and 3).
